# Ultra-High-Frequency Ultrasound Mapping of the Superficial Circumflex Iliac and Superficial Inferior Epigastric Vessels: An Anatomical Study

**DOI:** 10.3390/diagnostics15101210

**Published:** 2025-05-11

**Authors:** Spencer Chia-Hao Kuo, Ryo Karakawa, Hirofumi Imai, Shintaro Kagimoto, Yukio Seki, Nobuko Suesada, Hidehiko Yoshimatsu, Tomoyuki Yano

**Affiliations:** 1Department of Plastic and Reconstructive Surgery, Kaohsiung Chang Gung Memorial Hospital and Chang Gung University College of Medicine, Kaohsiung 83301, Taiwan; spenc19900603@gmail.com; 2Department of Plastic and Reconstructive Surgery, Cancer Institute Hospital of the Japanese Foundation for Cancer Research, Tokyo 135-8550, Japan; hirofumi.imai@jfcr.or.jp (H.I.); shintaro.kagimoto@jfcr.or.jp (S.K.); yukio.seki@jfcr.or.jp (Y.S.); nobuko.suesada@jfcr.or.jp (N.S.); hidehiko.yoshimatsu@gmail.com (H.Y.); yanoprs@icould.com (T.Y.)

**Keywords:** SIEA flap, SCIP flap, microsurgery, ultra-high-frequency ultrasound

## Abstract

**Background:** The superficial vessel system in the lower abdomen, including the superficial circumflex iliac artery (SCIA) and superficial inferior epigastric artery (SIEA), is widely used in reconstructive microsurgery. Preoperative ultrasonography, particularly ultra-high-frequency ultrasound (UHFUS), enhances surgical planning by providing high-resolution imaging. This study aimed to utilize UHFUS to examine the SCIA, SCIV, SIEA, and SIEV for reconstructive surgery planning. **Methods:** This prospective study included 25 patients undergoing free DIEP flap breast reconstruction. Patients with horizontal lower abdominal scars were excluded. Preoperative UHFUS, using a 48 MHz transducer, was performed to map and measure the superficial branch of SCIA (sSCIA), SCIV, SIEA, and SIEV. The vessel location, diameter, depth, and course were documented and analyzed. **Results:** Twenty-five female patients (50 hemiabdomens) aged 41 to 66 were included. The mean BMI was 21.6 kg/m^2^ (range: 18.4–30.4 kg/m^2^). At the ASIS level, the mean diameter of the sSCIA, SIEA, SCIV, and SIEV were 0.76 mm, 0.63 mm, 1.72 mm, and 2.18 mm, respectively. A superior lateral pedicle course was observed in 98% of the sSCIA. All patients had at least one detectable superficial artery, with 96% showing detectable arteries on both sides of the lower abdomen. **Conclusions:** UHFUS effectively maps superficial vessels in the lower abdomen for reconstructive surgery. The SCIA and SCIV are reliably detectable, while the SIEA is less consistently identified. UHFUS enhances flap design by providing precise vessel localization and sizing, leading to safer and more efficient surgeries.

## 1. Introduction

The superficial vessel system of the lower abdomen has gained popularity and is now widely used in reconstructive microsurgery. The superficial circumflex iliac artery perforator (SCIP) flap is supplied by the superficial circumflex iliac artery (SCIA) and vein (SCIV), and has been applied in the reconstruction of the upper and lower extremities, head and neck, trunk, and even the breast [1,2,3,4,5,6,7,8,9]. The SCIA can arise from the femoral artery, the superficial femoral artery, the deep femoral artery, or the lateral circumflex artery. It generally runs parallel to the inguinal ligament and bifurcates into the superficial branch (sSCIA), which gives off perforators to the groin skin, and the deep branch (dSCIA) which gives off branches to the sartorius muscle, the iliac bone and perforators to the skin around the ASIS.

A similar flap from the lower abdomen can also be harvested based on the superficial inferior epigastric artery (SIEA) [10,11,12]. The SIEA can originate independently from the common femoral artery, or form a common trunk with the SCIA. It generally runs superiorly and medially, passing superficial to the inguinal ligament and traveling within the subcutaneous tissue of the lower abdominal wall. The superficial inferior epigastric vein (SIEV) has also been widely recognized as the main drainer of the superficial system dominated deep inferior epigastric perforator (DIEP) flap.

Ultrasonography has been widely utilized in the surgical planning of reconstructive microsurgery. Conventional high-frequency ultrasound devices typically operate at frequencies of 15–18 MHz, which are highly operator-dependent, making it challenging to obtain precise imaging of small anatomical structures [13]. The ultra-high-frequency ultrasound (UHFUS) system, with frequencies up to 48 MHz and even as high as 70 MHz, provides a resolution as fine as 30–50 μm, allowing for the more precise imaging of small structures. It is frequently used in the planning of lymphatic surgeries and in the detailed assessment of vasculature in reconstructive microsurgery [14].

The aim of this study was to utilize UHFUS in the anatomical examination of the sSCIA, SCIV, SIEA, and SIEV. With the aid of UHFUS, we are able to obtain high-resolution images of the lower abdominal vasculature, allowing for a comprehensive study that covers the location, diameter, depth, and course of the superficial vessels in the lower abdomen. 

## 2. Materials and Methods

This study was approved by the ethical review board of the Cancer Institute Hospital of the Japanese Foundation for Cancer Research. We conducted a prospective study involving patients who underwent free DIEP flap breast reconstruction between 1 January 2024 and 30 July 2024. Patients with horizontal scars over the lower abdomen were excluded from this study. UHFUS examinations were performed during the surgical design and marking process the day before the operation. Patient information, including age, sex, BMI, and smoking status, was retrieved from medical records.

All UHFUS exams were performed by the first author, S.C. Kuo, using the Vevo MD machine (FUJIFILM VisualSonics, Inc., Toronto, ON, Canada). On the day before surgery, the primary surgeon completed the DIEP flap design and recipient site design for all patients. Following the design, each patient was placed in a supine position and evaluated with the Vevo MD machine using a 48 MHz transducer. First, the vertical midline and the horizontal line at the level of the anterior superior iliac spine (ASIS) were marked. We then searched for the sSCIA, its vena comitans (VC), SCIV, SIEA, VC of SIEA, and SIEV at the ASIS level. The vessels were traced proximally and distally until they became too deep or too small to be captured with clear images. If neither the sSCIA nor the SIEA could be identified, a second staff member (R. Karakawa) would repeat the exam and verify the results. Images were captured for further measurements at the ASIS level and at the most proximal point possible (Figure 1A and Figure 2A). The locations of the vessels were marked with marking pens, and by connecting the most proximal point of each vessel to its location at the ASIS level, we were able to determine the vessel’s course. The course was classified as either superior-lateral, superior, or superior-medial. Finally, the vessel diameter was measured after completing the entire exam (Figure 1B and Figure 2B). All measurements were taken three times and then averaged to minimize bias. All images with mapped vessels were processed using Image J v.1.4 (National Institutes of Health, Bethesda, MD, USA) to measure the distances from the vessels to the ASIS and the midline (Figure 3). Data were processed and analyzed using SPSS v.22 statistical software (IBM, Armonk, NY, USA). The normality of the data was evaluated using the Shapiro–Wilk test, and the data were presented as mean ± standard deviation (SD) or median ± interquartile range (IQR) accordingly.

## 3. Results

Table 1 summarizes the general information of the patients included in this study. A total of 25 patients, representing 50 hemiabdomens, were included. The patients ranged in age from 41 to 66 years, and all were female. The mean BMI was 21.6 kg/m^2^, with a range from 18.4 kg/m^2^ to 30.4 kg/m^2^. Five patients (20%) were smokers. All patients had at least one detectable superficial artery in the lower abdomen, and 96% of the patients had at least one detectable superficial artery on both sides of the lower abdomen.

Table 2 shows the anatomical characteristics of the superficial branch of the superficial circumflex iliac artery, the VCs, and the superficial circumflex iliac vein, as measured using UHFUS. The sSCIA was detectable in 48 hemiabdomens (96%). The average diameter of the vessel at the ASIS level and when traced proximally was 0.76 ± 0.25 mm and 0.97 ± 0.25 mm, respectively. The average depth of the vessel at the ASIS level was 5.42 ± 2.13 mm (Figure 4A). Correlation analysis between BMI and the sSCIA depth at the ASIS level showed a weak to moderate positive correlation: as BMI increases, the vessel depth tends to increase (Spearman’s rho (ρ) = 0.306; *p*-value = 0.046). A total of 43 VC (86%) were detected running alongside the sSCIA, with an average diameter of 0.94 ± 0.37 mm at the ASIS level and 1.34 ± 0.58 mm when traced proximally. The average distance from the sSCIA to the ASIS (“hot zone”) and to the midline was 18.71 ± 15.44 mm and 91.02 ± 15.89 mm, respectively, at the ASIS level. Most sSCIAs (98%) followed a superior lateral pedicle course. At least one dominant SCIV was detectable in 48 hemiabdomens (96%). The average diameter of the SCIV at the ASIS level and when traced proximally was 1.72 ± 0.60 mm and 2.07 ± 0.48 mm, respectively. The average depth of the SCIV at the ASIS level was 3.66 ± 3.29 mm (Figure 4B). Correlation analysis between BMI and the SCIV depth at the ASIS level also showed a weak to moderate positive correlation (Spearman’s rho (ρ) = 0.319; *p*-value = 0.027). The average distance between the SCIV and the ASIS (“hot zone”) was 10.46 ± 14.71 mm, and the distance between the sSCIA and the SCIV was 15.47 ± 19.10 mm. All SCIVs followed a superior lateral pedicle course.

Table 3 shows the anatomical characteristics of the superficial inferior epigastric artery, the VCs, and the superficial inferior epigastric vein, as measured using UHFUS. The SIEA was detectable at the ASIS level in only 11 hemiabdomens (22%). The diameter of the vessel at the ASIS level and when traced proximally was 0.63 ± 0.22 mm and 1.05 ± 0.20 mm, respectively. The average depth of the vessel at the ASIS level was 4.94 ± 3.60 mm. Eight accompanying veins were detected running alongside the SIEA, with an average diameter of 0.64 ± 0.28 mm at the ASIS level and 1.35 ± 1.06 mm when traced proximally. The average distance from the SIEA to the ASIS and to the midline was 51.32 ± 20.57 mm and 50.92 ± 22.26 mm, respectively, at the ASIS level. About half of the SIEAs (6/11, 55%) followed a superior medial pedicle course. At least one dominant SIEV was detectable in all hemiabdomens. The average diameter of the SIEV at the ASIS level and when traced proximally was 2.18 ± 0.48 mm and 2.39 ± 0.57 mm, respectively. The average depth of the vessel at the ASIS level was 3.46 ± 2.80 mm (Figure 4C). Correlation analysis between BMI and the SIEV depth at ASIS level showed a moderate positive correlation (Spearman’s rho (ρ) = 0.375; *p*-value = 0.007). The average distance between the SIEV and the midline (“hot zone”) was 47.94 ± 10.11 mm. A total of 38 out of 50 SIEVs (76%) followed a superior medial pedicle course.

The vessel diameter was compared between smokers and non-smokers (Table 4). The *p*-values were 0.78 and 0.96, respectively, for sSCIA at the ASIS level and proximal. For SIEA at the ASIS level and proximal, the *p*-values were 0.08 and 0.93, respectively. No significant difference was found between smokers and non-smokers.

## 4. Discussion

The first vascularized iliac bone flap based on the SCIA was reported by Taylor and Watson in 1978 [15]. In 2004, Koshima et al. introduced the SCIP flap, which has gradually become more commonly used in reconstructive microsurgery [7,8,16,17,18,19]. According to anatomical studies based on intraoperative findings and computed tomographic angiography (CTA), the SCIA arises from the femoral artery, the superficial femoral artery, the deep femoral artery, or the lateral circumflex artery [20]. It then bifurcates into the sSCIA, which gives off perforators to the groin skin, and the deep branch (dSCIA), which gives off branches to the sartorius muscle, the iliac bone and perforators to the skin around the ASIS [20,21,22]. A cadaveric dissection study published by Yoshimatsu et al. suggested that the bifurcation point of the sSCIA and the dSCIA could be identified within 2 cm of a fixed site—6 cm from the pubic tubercle to the ASIS and 3 cm caudal from that point—in 85% of the specimens [16]. In the current study, the “hot zone” for locating the sSCIA pedicle is 3.27 to 34.15 mm medial to the ASIS (Table 2). Today, the SCIP flap, whether based on the sSCIA, the dSCIA, or both superficial and deep branches in a chimeric design, is widely used in the reconstruction of various anatomical regions.

A similar flap from the lower abdomen can also be harvested based on the SIEA, commonly referred to as the SIEA flap [10,11,12]. Compared to the workhorse DIEP flap for breast reconstruction, the SIEA flap or the SIEA-SCIA combined flap spares the fascia and rectus muscle dissection, leaving only an abdominoplasty scar, which minimizes donor site morbidity. An anatomical study of the SIEA system, based on CTA studies published by Kita et al., confirmed that the SIEA was present in 92.4% of the hemiabdomens they reviewed [23]. It originated independently from the common femoral artery in 59.7% of the hemiabdomens, and it formed a common trunk with the SCIA in 32.6% of the cases. The SIEV was present in 99.3% of the hemiabdomens [23]. In the current study, the SIEA was detectable by Vevo MD at the ASIS level in only 22% of the hemiabdomens. This detection rate differs from the previous CTA study published by Kita et al., likely due to the design of Vevo MD, which is optimized for examining superficial body structures. As a result, some SIEAs may be too deep to be detected at the ASIS level. In contrast, CTA studies trace the vessels starting from their point of origin. Additionally, the examination was conducted while the patient was awake in the outpatient clinic, so the examination time was not unlimited. Although the result differs from that of the prior CTA study, we believe this number reflects the real-time condition of living Asian women in the clinical setting, which is a strength of this prospective study on living human subjects.

In the current study, the sSCIA was detectable by Vevo MD at the ASIS level in 96% of the hemiabdomens, and at least one set of detectable superficial arteries and veins was found in the lower abdomen of all patients. The superficial arteries detected at the ASIS level typically had an average diameter of over 0.5 mm, which, based on our experience, often measures greater than 1 mm when dissected proximally near their origin. With the help of Vevo MD, we can map out sizable superficial arteries and include one of the arteries in our flap. On the other hand, venous drainage is often the primary concern when planning a flap based on the superficial vessel system of the lower abdomen. Since the average distance between the dominant artery and vein is usually not large, we tend to include and position the dominant superficial vein at the center of the flap. Although there is ongoing debate regarding the necessity and value of preoperative imaging, we believe that this methodology can provide additional guidance when planning an sSCIA-based or SIEA-based flap.

One specific strength of our study is its prospective design and the use of the UHFUS system on living human subjects, which allowed us to generate the “hot zone” for detecting vessels at the ASIS level based on these data. Ultrasonography holds an indispensable role as it provides real-time evaluation, allowing us to assess the vessels directly on the patient in different positions. The UHFUS enhances this capability by offering a high resolution, which enables the more precise measurement of vessel diameter. However, there are some limitations to this study. First and foremost, the absence of a comparison with a gold-standard modality such as computed tomography angiography limits our ability to evaluate diagnostic performance metrics, including sensitivity, specificity, the positive predictive value, negative predictive value, and overall accuracy. Future studies should aim to address the diagnostic accuracy of UHFUS through a direct comparison with established imaging techniques. Second is the operator-dependent nature of ultrasonography, which can introduce some measurement bias. To mitigate this, we repeated each measurement three times using the Vevo MD machine. Thirdly, the Vevo MD is designed to examine superficial body structures, making precise diameter measurements for structures deeper than 20 mm somewhat challenging. Tiny structures located deeper than 25–30 mm may go undetected. Therefore, vessel evaluation may become more challenging in very obese patients. Lastly, this is a single-center study with a study population limited to a single race and gender. Further studies with larger and more diverse populations are needed to obtain more comprehensive data.

## 5. Conclusions

In conclusion, all patients in the current study had at least one sizable set of superficial arteries and veins in the lower abdomen. When a free lower abdominal skin flap transfer is indicated, preoperative ultrasonography enables the precise mapping of the flap’s pedicle vessels, confirmation of the pedicle diameter, and the design of a well-vascularized flap. This approach simplifies flap harvest and enhances patient safety during free flap transfer surgery.

## Figures and Tables

**Figure 1 diagnostics-15-01210-f001:**
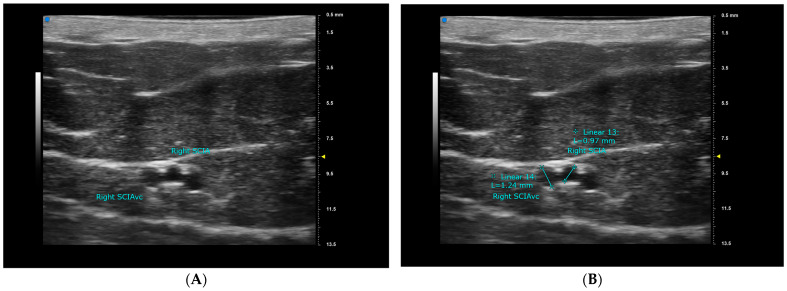
The images captured from Vevo MD demonstrating the superficial branch of the right superficial circumflex iliac artery (sSCIA) and its vena comitans (VC) at the anterior superior iliac spine (ASIS) level for case number 25 (**A**). Measurements were performed on the captured image after the whole ultrasound exam process was complete (**B**).

**Figure 2 diagnostics-15-01210-f002:**
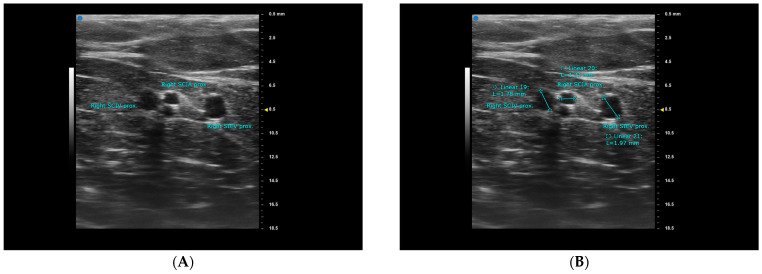
The images captured from Vevo MD demonstrating the superficial branch of the right superficial circumflex iliac artery (sSCIA) and its vena comitans (VC) traced to the most proximal level possible for case number 25. The most proximal level possible for the right superficial inferior epigastric vein (SIEV) was also captured in the same image (**A**). Measurements were performed on the captured image after the whole ultrasound exam process was complete (**B**).

**Figure 3 diagnostics-15-01210-f003:**
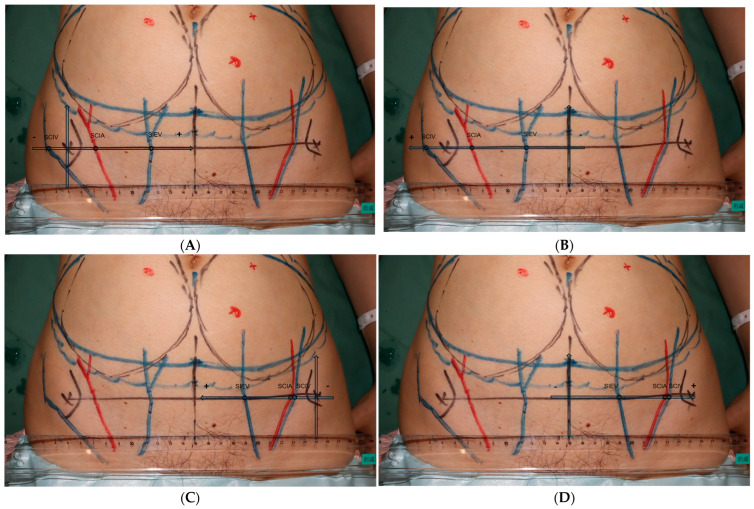
The figures demonstrate the measurement of the distances from the vessels to the anterior superior iliac spine (ASIS) (**A**,**C**), and from the vessels to the midline (**B**,**D**) for case number 24. SCIA = superficial circumflex iliac artery; SCIV = superficial circumflex iliac vein; SIEA = superficial inferior epigastric artery; SIEV = superficial inferior epigastric vein.

**Figure 4 diagnostics-15-01210-f004:**
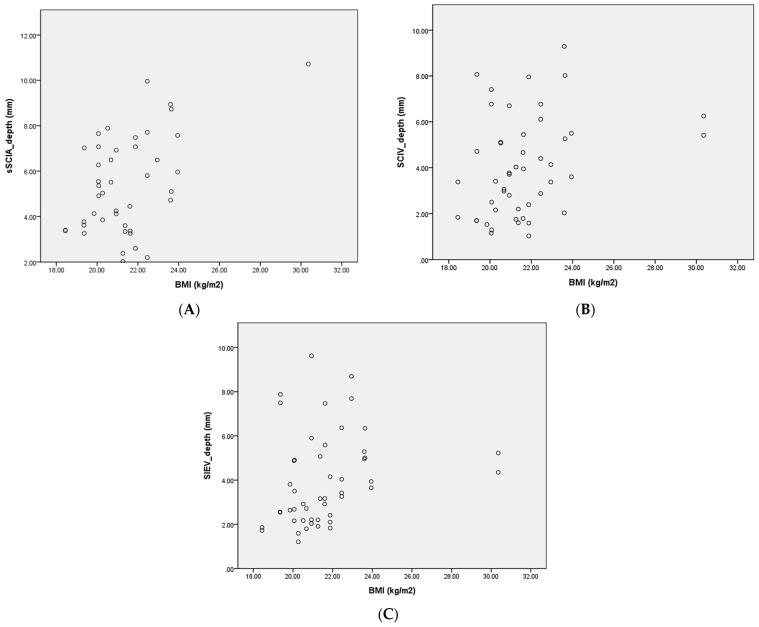
The relationship between body mass index (BMI) and the depth of the superficial branch of the superficial circumflex iliac artery (SCIA) (**A**), the superficial circumflex iliac vein (SCIV) (**B**), and the superficial inferior epigastric vein (SIEV) (**C**) at the anterior superior iliac spine (ASIS) level, presented in scattered dot plot.

**Table 1 diagnostics-15-01210-t001:** Demographic data of patients included in the current study.

Patient number	25
Hemiabdomens	50
age (mean)	51.1
Range	41−66
Sex-female	25
BMI (mean)	21.6
Range	18.4−30.4
Hemiabdomen width (mm, from midline to ASIS)	109.65 ± 7.50 *
Smoker	5/25, 20%
No detectable superficical arteries on both sides	0/25, 0%
Detectable SCIA or SIEA on one side only	1/25, 4%
Detectable SCIA or SIEA on both sides	24/25, 96%
Detectable SCIA and SIEA on one side	8/25, 32%
Detectable SCIA and SIEA on both sides	1/25, 4%

ASIS = anterior superior iliac spine; BMI = body mass index; SCIA = superficial circumflex iliac artery; SIEA = superficial inferior epigastric artery. * Presented as mean ± standard deviation.

**Table 2 diagnostics-15-01210-t002:** Statistical analysis of superficial branch of superficial circumflex iliac artery and vein measurements using the ultra-high-frequency Ultrasound (UHFUS) System.

	sSCIA	SCIA VC	SCIV
detectable at ASIS level	48/50, 96%	43/50, 86%	48/50, 96%
Average diameter at ASIS (mean ± SD, mm)	0.76 ± 0.25	0.94 ± 0.37	1.72 ± 0.60
Average depth at ASIS (mean ± SD, mm)	5.42 ± 2.13	5.42 ± 2.13	3.66 ± 3.29 *
Average diameter when traced proximal (mean ± SD, mm)	0.97 ± 0.25	1.34 ± 0.58 *	2.07 ± 0.48 *
Average distance to midline (mean ± SD, mm)	91.02 ± 15.89		101.58 ± 17.95
Average distance to ASIS (mean ± SD, mm)	18.71 ± 15.44 (3.27−34.15 ^ǂ^)		10.46 ± 14.71 * (−4.25–25.17 ^ǂ^)
Superior-lateral pedicle course	47/48, 98%		48/48, 100%
Superior pedicle course	1/48, 2%		0/48, 0%
Superior-medial pedicle course	0/48, 0%		0/48, 0%
Average SCIA-SCIV distance (mm)	15.47 ± 19.10 *		

* Presented as median ± interquartile Range. ^ǂ^ “Hot Zone” for vessel detection. ASIS = anterior superior iliac spine; sSCIA = superficial branch of superficial circumflex iliac artery; SCIV = superficial circumflex iliac vein; SD = standard deviation; VC = concomitant vein.

**Table 3 diagnostics-15-01210-t003:** Statistical analysis of superficial inferior epigastric artery and vein measurements using the ultra-high-frequency Ultrasound (UHFUS) System.

	SIEA	SIEA VC	SIEV
detectable at ASIS level	11/50, 22%	8/50, 16%	50/50, 100%
Average diameter at ASIS (mean ± SD, mm)	0.63 ± 0.22 *	0.64 ± 0.28 *	2.18 ± 0.48
Average depth at ASIS (mean ± SD, mm)	4.94 ± 3.60 *	4.94 ± 3.60 *	3.46 ± 2.80 *
Average diameter when traced proximal (mean ± SD, mm)	1.05 ± 0.20 *	1.35 ± 1.06 *	2.39 ± 0.57
Average distance to midline (mean ± SD, mm)	50.92 ± 22.26 *		47.94 ± 10.11 (37.83−58.05 ^ǂ^)
Average distance to ASIS (mean ± SD, mm)	51.32 ± 20.57 *		62.63 ± 12.44
Superior-lateral pedicle course	2/11, 18%		2/50, 4%
Superior pedicle course	3/11, 27%		10/50, 20%
Superior-medial pedicle course	6/11, 55%		38/50, 76%
Average SIEA-SIEV distance	13.65 ± 14.88 *		

* Presented as median ± interquartile Range. ^ǂ^ “Hot Zone” for vessel detection. ASIS = anterior superior iliac spine; SIEA = superficial inferior epigastric artery; SIEV = superficial inferior epigastric vein; SD = standard deviation; VC = concomitant vein.

**Table 4 diagnostics-15-01210-t004:** Vessel diameter comparison between smokers and non-smokers using the Mann–Whitney U test.

Vessel Diameter (mm) *	Smoker (5 Patients)	Non-Smoker (20 Patients)	*p* Value (Mann-Whitney U Test)
sSCIA, ASIS	0.63 ± 0.44	0.74 ± 0.32	0.78
sSCIA, proximal	0.90 ± 0.29	0.96 ± 0.45	0.96
SCIV, ASIS	1.97 ± 0.39	1.62 ± 1.03	0.35
SCIV, proximal	2.16 ± 0.28	2.01 ± 0.47	0.24
SIEA, ASIS	0.53 ± 0.12	0.71 ± 0.14	0.08
SIEA, proximal	1.08 ± 0.21	1.04 ± 0.14	0.93
SIEV, ASIS	2.30 ± 0.72	2.17 ± 0.79	0.25
SIEV, proximal	2.53 ± 0.65	2.25 ± 0.83	0.14

* Presented as median ± interquartile Range. ASIS = anterior superior iliac spine; sSCIA = superficial branch of superficial circumflex iliac artery; SCIV = superficial circumflex iliac vein; SIEA = superficial inferior epigastric artery; SIEV = superficial inferior epigastric vein.

## Data Availability

The data presented in this study are available upon request from the corresponding author. The data are not publicly available due to privacy and ethical restrictions.

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
