# Peer review of "Ultra-High-Frequency Ultrasound Mapping of the Superficial Circumflex Iliac and Superficial Inferior Epigastric Vessels: An Anatomical Study"

_diagnostics, 2025, doi:10.3390/diagnostics15101210_

Round 1
Reviewer 1 Report
Comments and Suggestions for Authors
The authors had an interesting idea to study the anatomy of superficial vasculature of the lower abdomen (SCIA, SCIV, SIEA, SIEV) using an ultra-high-frequency-ultrasound (UHFUS) of 48 MHz and its clinical impact in reconstructive microsurgery. It was a real pleasure to go through the different sections starting from Introduction’s general information which provide interesting details for the topic preparing the reader with that will ensue.
In the upcoming paragraphs it was presented and analyzed the methodology of the study with author’s proposed measurements (distance from ASIS horizontal line and vertical midline, vessels’ size with their diameter distally and proximally). The location and the course of the examined vessels were marked with marked pens. Additionally, all measurements were taken three times and averaged trying to minimize bias. Shapiro-Wilk test used for the evaluation of data’s’ normality.
In the following sections their results were thoroughly developed within four Tables that summarized the statistical analysis of the findings. An additional fifth Table, a scattered dot plot, completes the findings showing the relationship between BMI and the depth of the studied vessels at the ASIS line. It will certainly provide substantial assistance in the clinical field. Finally, in their discussion the findings were adequately presented and analyzed along with a last paragraph in which the limitations of the study were thoroughly considered.
Generally, the manuscript was correctly “build” covering all aspects of subject.
We think that this article deserves to be published.
Author Response
We sincerely thank the reviewer for your thoughtful and encouraging comments on our manuscript. We are very pleased to hear that the anatomical approach and clinical implications of our study using ultra-high-frequency ultrasound (UHFUS) were appreciated. We are especially grateful for the recognition of our methodology, data presentation, and discussion of the study's limitations. Your supportive feedback is highly motivating, and we truly appreciate your recommendation for publication.
Reviewer 2 Report
Comments and Suggestions for Authors
"An Anatomical Study of the Superficial Branch of Superficial Circumflex Iliac Vessels and the Superficial Inferior Epigastric Vessels Using the Ultra-High Frequency Ultrasound System"
- Title
- The current title is too long and slightly redundant.
- Suggested title "Ultra-High Frequency Ultrasound Mapping of the Superficial Circumflex Iliac and Superficial Inferior Epigastric Vessels: An Anatomical Study"
- Introduction
- The formatting on page 2 is not compliant with MDPI layout guidelines—the body text exceeds the right margin.
- While the clinical applications of the vessels are well emphasized, there is no description of their basic anatomy. For clarity and completeness, please include:
- Origin and branching patterns
- Number and distribution of perforators (lateral vs. medial)
- Typical trajectory
- Relation with surrounding structures.
- Including this information will make the study more accessible to readers without a deep background in flap anatomy.
- Materials and Methods
- Well-structured and clearly presented. No major revisions needed.
- Results
4.1 Consider dividing the Results section into subsections for clarity, such as:
- 3.1 Statistical Results for the Superficial Circumflex Iliac Artery and Vein
- 3.2 Statistical Results for the Superficial Inferior Epigastric Artery and Vein
4.2 Throughout the text and tables, the term “size” is used to describe vessel caliber. However, this term is too broad. Please replace it with “diameter”, as you are referring specifically to cross-sectional diameter (Line 121–122 and in Tables 2 & 3, and other places throughout the manuscript).
4.3 In the analysis comparing smokers and non-smokers, you mention that no significant differences were found. Please include the exact p-values in the main text to support this conclusion, not only in the table.
4.4 Consider performing a correlation analysis between BMI and vessel depth (as illustrated in Figure 5) to provide statistical support. A Spearman’s rho test would be appropriate given the data distribution.
- Discussion
5.1 A brief discussion on how UHFUS findings can influence flap design or planning (e.g., choice of dominant perforator, flap orientation, inclusion of superficial veins) would further strengthen the clinical relevance of your findings.
5.2 The first paragraph of the Discussion (lines 174–184), which outlines the historical and anatomical background of the SCIP flap, may be better placed in the Introduction section. This would balance the manuscript and allow for more focused analysis in the Discussion.
Overall Assessment
This manuscript addresses an important topic in reconstructive microsurgery using a novel high-resolution imaging modality. The study is well-executed and presents valuable clinical data. With the major revisions above, particularly regarding anatomical background, clarity in statistical terminology, and structure in the Results and Discussion sections, the manuscript will be significantly improved.
Good luck!
Author Response
- Title
The current title is too long and slightly redundant. Suggested title "Ultra-High Frequency Ultrasound Mapping of the Superficial Circumflex Iliac and Superficial Inferior Epigastric Vessels: An Anatomical Study"
- We agree this is a better title for the article. This modification has been made in the revised manuscript in RED. Thanks for your professional comment!
- Introduction
The formatting on page 2 is not compliant with MDPI layout guidelines—the body text exceeds the right margin.
- We will improve this with the revised manuscript. Thanks for your professional comment!
While the clinical applications of the vessels are well emphasized, there is no description of their basic anatomy. For clarity and completeness, please include:
- Origin and branching patterns
- Number and distribution of perforators (lateral vs. medial)
- Typical trajectory
- Relation with surrounding structures.
Including this information will make the study more accessible to readers without a deep background in flap anatomy.
- Yes, we have included this important information in the revised manuscript, highlighted in RED text within the Introduction section. Thank you for your professional and valuable comment!
- Materials and Methods
Well-structured and clearly presented. No major revisions needed.
- Thanks for your professional comment!
- Results
4.1 Consider dividing the Results section into subsections for clarity, such as:
- 3.1 Statistical Results for the Superficial Circumflex Iliac Artery and Vein
- 3.2 Statistical Results for the Superficial Inferior Epigastric Artery and Vein
- Yes, we agree that the statistical results for the SCIA/V and the statistical results for the SIEA/V should be divided for clarity. Therefore, the information regarding the SCIA/V and the SIEA/V are separated into two paragraphs (paragraphs 2 and 3 in the result section). Thanks for your professional comment!
4.2 Throughout the text and tables, the term “size” is used to describe vessel caliber. However, this term is too broad. Please replace it with “diameter”, as you are referring specifically to cross-sectional diameter (Line 121–122 and in Tables 2 & 3, and other places throughout the manuscript).
- Yes, we agree that “diameter” is a better term than “size”. We will make the replacement in the revised manuscript (RED text). Thanks for your professional comment!
4.3 In the analysis comparing smokers and non-smokers, you mention that no significant differences were found. Please include the exact p-values in the main text to support this conclusion, not only in the table.
- Yes, we have added the exact p-values in the revised manuscript in RED. However, to avoid presenting a list-like account of p-values, we chose to report only the important data for the superficial branch of the SCIA and the SIEA. Thanks for your professional comment!
4.4 Consider performing a correlation analysis between BMI and vessel depth (as illustrated in Figure 5) to provide statistical support. A Spearman’s rho test would be appropriate given the data distribution.
- Yes, we have performed the correlation analysis and added the Spearman's rho and the p-values in the revised manuscript in RED (Second and third paragraph in the result section). Thanks for your professional comment!
- Discussion
5.1 A brief discussion on how UHFUS findings can influence flap design or planning (e.g., choice of dominant perforator, flap orientation, inclusion of superficial veins) would further strengthen the clinical relevance of your findings.
- Yes, we have added descriptions in terms of flap planning using the UHFUS in the revised manuscript in RED (Third paragraph in the Discussion section). Thanks for your professional comment!
5.2 The first paragraph of the Discussion (lines 174–184), which outlines the historical and anatomical background of the SCIP flap, may be better placed in the Introduction section. This would balance the manuscript and allow for more focused analysis in the Discussion.
- Thank you for your insightful comment. While we believe the historical background is more suitably addressed in the Discussion section, we have expanded the anatomical description of the vessels in the Introduction as per your suggestion. Thanks for your professional comment!
Overall Assessment
This manuscript addresses an important topic in reconstructive microsurgery using a novel high-resolution imaging modality. The study is well-executed and presents valuable clinical data. With the major revisions above, particularly regarding anatomical background, clarity in statistical terminology, and structure in the Results and Discussion sections, the manuscript will be significantly improved.
Good luck!
- We sincerely thank the reviewer for the thoughtful and encouraging feedback on our manuscript. We remain happy to make further revisions if needed.
Reviewer 3 Report
Comments and Suggestions for Authors
The novelty of this study lies in the use of UHFUS for assessing the lower abdominal superficial vessel system in planning free flaps. This is particularly interesting, and the authors should be commended for promoting the use of novel technologies to aid microsurgery, especially when using non-invasive modalities and empowering the surgeons with a tool they can use themselves. However, despite the prospective nature of the study, I believe that there are significant limitations which significantly affect the utility of your findings. While I am convinced that UHFUS is going to be an invaluable tool for plastic surgeons moving forward, I do not believe that your study demonstrates this.
I would have expected your study to assess the viability of perforator vessels to be used for raising SCIP or SIEA flaps. You could have used UHFUS to also map perforator vessels from the DIEA as well. Instead you used the setting of patients requiring DIEP flaps to assess the superficial vascular system of the inferior abdomen. To demonstrate the effectiveness of a novel diagnostic tool, the following should be performed: a comparison to other established or “gold standard” methods. For DIEP flap surgery, this is arguably considered to be the CT angio. As such you should perform calculation of sensitivity (ability to correctly identify true positives, i.e. viable perforators or SIEV/SIEA), specificity (ability to correctly identify true negatives, i.e. inadequate perforators), Positive Predictive Value (PPV), Negative Predictive Value (NPV), Accuracy (i.e. the proportion of correct classifications overall), a ROC (Receiver Operating Characteristic) Curve and AUC (Area Under the Curve) to summarize diagnostic performance and overall classification thresholds. Because of your study design, no comparisons can be made. Therefore, you should at least provide the assessment of an agreement rate between preoperative assessment using UHFUS and intraoperative findings. While it may be challenging to restructure this research project you should address all said limitations in a coherent manner.
Aside from this main criticism to your work, please find below a list of minor comments and suggestions which I hope can also be addressed:
- The higher the frequency the lower the depth which by your admission means that resolution is acceptable for 25-30 mm of depth. What happens in patients with a thick abdominal subcutis which are usually considered prime candidates for DIEP flaps? What about obese patients? This should be addressed as a possible relative limitation.
- Other than vessel diameter and location, can this methodology be used to assess perfusion and flow velocity? Can the system be used intraoperatively to aid surgeons?
- Can you infer whether there is vascular dominance of the superficial venous system through your system?
- Some surgeons perform DIEP flaps in a “free style” manner only using a handheld doppler probe, without increasing surgery duration or complication rates. [PMID: 36847143] This is quite contentious as different schools of thought are adamant supporters of the benefits of preoperative imaging. Regardless of your stance on the matter, I believe this point of contention to be especially relevant to your manuscript and should be addressed.
- Your manuscript does not focus on the cost-effectiveness of UHFUS, however the relevant costs cannot be ignored, with the equipment alone ranging from 80,000-200,000+ Euros. A basic color doppler US system will cost anywhere between 10,000-30,000 Euros and could come close to doing the same job. How would you justify this very high price tag when the same procedure is successfully performed without any of the ultraspecialized equipment which is not at every plastic surgery department’s reach?
- I believe that other uses make UHFUS an exceptional tool, such as lymphatics or smaller caliber vessels. [PMID: 36294872] Please provide your thoughts and implement in the discussion.
Author Response
We sincerely thank the reviewers for their thoughtful and detailed comments on our manuscript. We are encouraged by the recognition of the novelty of using ultra-high-frequency ultrasound (UHFUS) to assess the superficial vascular system of the lower abdomen and its potential utility in free flap planning. As noted, our goal was to explore the feasibility of implementing a surgeon-operated, non-invasive imaging modality in microsurgical planning.
We fully acknowledge the limitations raised regarding our current study design, particularly the absence of a comparison with a gold-standard modality such as computed tomography angiography (CTA). Our primary objective was to investigate the anatomical characteristics of the SCIA, SIEA, and associated superficial venous systems using UHFUS. We hope that this work will contribute to better preoperative planning of lower abdominal superficial vessel-based flaps, including the SCIP flap, SIEA flap, and even SCIP-SIEA combined flaps when needed. However, due to the limited number of SCIP flap cases at our institution, we conducted this study in patients undergoing DIEP flap breast reconstruction, which is routinely performed in our practice. We agree that the inclusion of CTA would have strengthened the study and provided additional valuable data.
Although the prospective design of our study limited the scope to superficial vessel mapping, we recognize the importance of evaluating diagnostic performance metrics such as sensitivity, specificity, positive predictive value (PPV), negative predictive value (NPV), and overall accuracy. We also agree that the inclusion of a receiver operating characteristic (ROC) curve and calculation of the area under the curve (AUC) would be essential in future validation studies.
In response to the reviewer’s suggestions, we have now expanded the discussion of the study’s limitations in the revised manuscript (in GREEN text, in the 4th paragraph in the Discussion section). We specifically highlight the absence of a reference standard for comparison and emphasize the need for future studies designed to assess the diagnostic accuracy of UHFUS. We sincerely appreciate the reviewer’s insightful feedback, which has helped us better define the scope and limitations of our work and will guide the direction of future research to establish the clinical value of UHFUS in microsurgical flap planning
Comment1: The higher the frequency the lower the depth which by your admission means that resolution is acceptable for 25-30 mm of depth. What happens in patients with a thick abdominal subcutis which are usually considered prime candidates for DIEP flaps? What about obese patients? This should be addressed as a possible relative limitation.
Response: Yes, thank you for your insightful comment. The primary objective of this study was to map and characterize the SCIA/V and SIEA/V, with the aim of facilitating the planning of SCIP flaps, SIEA flaps, and even SCIP-SIEA combined flaps in the future. In obese patients, these vessels can generally still be visualized in the groin region using UHFUS. However, in cases of extreme obesity—which are relatively uncommon in the Asian population—vessel evaluation may become more challenging. We have now included this point in the limitations section of the manuscript (highlighted in GREEN). We appreciate your valuable feedback.
Comment2: Other than vessel diameter and location, can this methodology be used to assess perfusion and flow velocity? Can the system be used intraoperatively to aid surgeons?
Response: Yes, thank you for the thoughtful question. While this methodology allows for the assessment of flow velocity within the vessel, it is important to note that flow velocity alone does not always correlate with overall flap perfusion. Flap perfusion is more accurately evaluated using modalities such as indocyanine green (ICG) angiography. However, the UHFUS system can be utilized intraoperatively and holds potential as a real-time, surgeon-operated tool to assist in decision-making during flap surgery. We appreciate your valuable feedback.
Comment3: Can you infer whether there is vascular dominance of the superficial venous system through your system?
Response: Yes, thank you for the very professional comment. While this methodology enables the measurement of superficial vessel diameter, we believe it is challenging to infer vascular dominance in a DIEP flap of the superficial venous system based solely on UHFUS findings. Intraoperative assessment using temporary vessel clamping to occlude the SIEV and evaluate flap circulation remains the most reliable method for determining venous dominance in a DIEP flap. We appreciate your valuable feedback.
Comment4: Some surgeons perform DIEP flaps in a “free style” manner only using a handheld doppler probe, without increasing surgery duration or complication rates. [PMID: 36847143] This is quite contentious as different schools of thought are adamant supporters of the benefits of preoperative imaging. Regardless of your stance on the matter, I believe this point of contention to be especially relevant to your manuscript and should be addressed.
Response: Thank you for your thoughtful comment. We acknowledge the ongoing debate regarding the necessity and value of preoperative imaging in DIEP flap surgery. As the reviewer noted, some experienced surgeons are able to perform DIEP flaps using a "free style" approach with only a handheld Doppler, without compromising operative time or complication rates [PMID: 36847143]. However, we believe that preoperative imaging—especially with emerging modalities such as UHFUS—can offer additional anatomical detail and guidance, particularly for less experienced surgeons planning a SCIP flap or a SIEA flap. We agree that this point of contention is highly relevant to the current manuscript, and we have now included a brief discussion on this topic to acknowledge differing surgical philosophies and to place our findings in a broader clinical context. (3rd Paragraph in the discussion section, in GREEN text)
Comment5: Your manuscript does not focus on the cost-effectiveness of UHFUS, however the relevant costs cannot be ignored, with the equipment alone ranging from 80,000-200,000+ Euros. A basic color doppler US system will cost anywhere between 10,000-30,000 Euros and could come close to doing the same job. How would you justify this very high price tag when the same procedure is successfully performed without any of the ultraspecialized equipment which is not at every plastic surgery department’s reach?
Response: Yes, thank you for your thoughtful comment. We acknowledge that a standard color Doppler ultrasound system is generally more cost-effective in the context of DIEP flap transfer. However, when planning an sSCIA-based or SIEA-based flap, the use of UHFUS can significantly enhance surgical planning, as these vessels are typically smaller and more challenging to visualize—particularly in the Asian population.
Comment6: I believe that other uses make UHFUS an exceptional tool, such as lymphatics or smaller caliber vessels. [PMID: 36294872] Please provide your thoughts and implement in the discussion.
Response: Thank you for this excellent suggestion. We fully agree that UHFUS holds significant potential beyond the scope of this study, particularly in the evaluation of lymphatic structures and small-caliber vessels. We have incorporated this important perspective into the third paragraph of the Introduction to emphasize the broader clinical applications of UHFUS in microsurgery and reconstructive planning. We sincerely appreciate your insightful comment!
Round 2
Reviewer 2 Report
Comments and Suggestions for Authors
The submitting authors made changes accordingly!
Good luck!
Reviewer 3 Report
Comments and Suggestions for Authors
Thank you for the detailed point by point response. I’m happy with the revisions in the current form.